# Multilevel negative binomial analysis of factors associated with numbers of antenatal care contacts in low and middle income countries: Findings from 59 nationally representative datasets

**Adugnaw Zeleke Alem** [1,2] *, **Biresaw Ayen Tegegne** [3], **Fantu Mamo Aragaw** [1], **Rediet Eristu Teklu** [1], **Tsegaw Amare Baykeda** [4,5]

1 Department of Epidemiology and Biostatistics, Institute of Public Health, College of Medicine and Health Sciences, University of Gondar, Gondar, Ethiopia, 2 Health Research Institute, Faculty of Health, University of Canberra, Bruce, Australia, 3 Department of Anesthesia, College of Medicine and Health Sciences, University of Gondar, Gondar, Ethiopia, 4 Department of Health Systems and Policy, Institute of Public Health, College of Medicine and Health Sciences, University of Gondar, Gondar, Ethiopia, 5 School of Public Health, The University of Queensland, Brisbane, Australia

* aduzeleke2201@gmail.com

**Data Availability Statement:** The data that support the findings of this study are available from the

## Abstract

### Background

Antenatal care (ANC) is one of the recommended interventions to reduce stillbirth, maternal, neonatal, and child mortality through early identification and management of pregnancy complications or pre-existing conditions. Although increasing number of ANC is a key priority of the 2016 WHO recommendations, ANC uptake in Low and Middle Income Countries (LMICs) is insufficient. Therefore, this study aimed to investigate factors associated with the number of ANC contacts in LMICs.

### Methods

Data for the study were drawn from 59 recent Demographic and Health Surveys (DHS) conducted in LMICS. We included a total sample of 520,377 mothers who gave birth in the five years preceding the survey. A multilevel negative binomial regression model was applied to identify factors that may affect number of ANC. Adjusted incidence rate ratios (AIRR) with 95% Confidence Interval (CI) were reported to show association.

### Results

This study found that mothers and their partner with higher educational attainment, mothers aged >35 years, mothers who had decision making autonomy, mothers from female headed household, mothers from richer and richest household, mothers exposed to media, and residing in urban areas had significantly more ANC contacts. However, number of ANC

Demographic and Health Surveys (DHS) Program. Redistribution of any DHS data directly or within any tool/dashboard is not permitted without written consent from DHS. Data are available at the DHS Program website (https://www.dhsprogram.com/data/dataset_admin/login_main.cfm).

**Funding:** The authors received no specific funding for this work.

**Competing interests:** The authors have declared that no competing interests exist.

**Abbreviations:** ANC, Antenatal Care; CI, Confidence Interval; DHS, Demographic and Health Survey; AIRR, Adjusted Incidence Rate Ratio; ICC, Intra-class Correlation Coefficient; LMICs, Low and Middle Income Countries; PCV, Proportional Change in Variance.

contacts were significantly lower among mothers who initiated ANC after 12 weeks of gestation and perceived healthcare access to be a big problem.

## Conclusion

Our results suggest that individual, household, and community-level factors were associated with number of ANC contacts among pregnant mothers in LMICs. Hence, local and international policymakers, and programmers should focus on improving community awareness about maternal health care services through mass media and outreach programs with especial emphasis on women's and their partners educational attainment, rural mothers, women's empowerment, and household socioeconomic status.

## Background

Antenatal care (ANC) is care provided to pregnant mothers by healthcare professionals to prevent pregnancy complications, provide labor counselling, and provide emergency preparedness to improve the health of both mother and newborns throughout pregnancy [1]. Annually, 210 million pregnancies occur, and about 140 million deliveries put maternal wellbeing and survival at risk [2, 3]. Even though recently launched Sustainable Development Goals (SDGs) aimed to reduce maternal mortality (MMR) to 70 per 100,000 live births by 2030 globally, MMR is a significant public health problem in low-income and middle-income countries (LMICs) [4, 5]. Worldwide, MMR reduced by 38% between 2000 and 2017, however, every day an estimated 810 women died from causes related to pregnancy [6]. Moreover, despite a 50% decline (from 36.6 to 18 deaths per 1000 live births) in the neonatal mortality rate between 1990 and 2017 globally, 1 in 37 neonates in sub–Saharan Africa (SSA) died [7, 8]. In 2019, an estimated 2.0 million stillbirths occur globally [9]. Of the all maternal deaths and stillbirths, 99% and 98% respectively occurred in LMICs [6, 7, 10]. This indicates an urgent need to tackle the root causes of still births and maternal and neonatal death in LMICs.

Antenatal care (ANC) is one of the recommended interventions to reduce stillbirths, maternal, newborn, and child mortalities through early identification and management of pregnancy complications or pre-existing conditions as well as educating women about the advantages of breastfeeding and the importance of family planning [1, 11, 12]. Increasing uptake of ANC had the key priority of the millennium development goal (MDG). During the MDG, ANC use increased from 25% to nearly 50%, but this was not enough [13].

Moreover, increasing uptake of ANC is the focus of the 2016 WHO recommendations on ANC for a positive pregnancy experience. This recommendation modified the minimum number of ANC contacts from the focused ANC model with a recommended minimum of four ANC visits (ANC4+) to eight contacts, with the first contact should be done within the first 12 weeks to reduce stillbirth, maternal, child and newborn deaths [14]. Higher quality ANC reduces the odds of a stillbirth by almost half and neonatal mortality by 39% [15, 16]. Besides, it is estimated that improving the coverage and quality of maternal healthcare could prevent 71% of neonatal deaths, 54% of maternal deaths and 33% of stillbirths in LMICs [17]. Various global and local studies identified that educational status of the women/partner, residence, parity, media exposure, age, wealth status, timing of first ANC, pregnancy intention, place of ANC visit, previous pregnancy complications, employment status/occupation, women's autonomy to decision, birth order, birth interval, marital status and distance from health facility were found to be associated factors of number of ANC/ANC4+/ANC8+ [18–28].

However, inconsistent results have been reported by different scholars. For example, a study conducted Azanaw MM, et.al, Fenta SM, et.al, and Tessema ZA, et.al revealed that women who are aged 15–24 years were less likely to receive ANC [20, 26, 29]. while a study by Hassen SS, et.al, and Ahinkorah BO, et.al observed that women who are aged 15–24 years were more likely to receive ANC [19, 25]. In addition, while women with their first child were less likely to receive ANC services by Azanaw MM, et al. [26], women with their first child were more likely to receive ANC services by Islam A, et al. [22]. These findings suggest that factors associated with ANC utilization are different in globally, which motivates us to conduct this study in LMICs by considering count nature ANC visits. Though there are studies conducted on the number of ANC and its associated factors in different countries in LMICs [18, 19, 21, 23–28], to the best of our knowledge, there is a scarcity of studies on the ANC contacts and its associated factors among pregnant women in LMICs based on the pooled national representative data. Moreover, there are studies conducted on the ANC utilization and its associated factors among women in sub-Saharan Africa and LMICs [20, 22, 30], however, these studies were focused on categorizing ANC contacts either ≥4 contacts or ≥8 contacts by using DHS conducted in different years. This study captures the count nature of ANC contacts to avoid loss of information and consider the time of DHS collection since some DHS surveys were conducted before 2016 WHO recommendations and some DHS surveys were conducted after 2016 WHO recommendations. Therefore, this study used multilevel negative binomial regression model to get a reliable estimate and provides sufficient information for studying the associated factors of ANC contacts.

## Methods

### Study population and data source

The study used data from the latest Demographic and Health Surveys (DHS) conducted between 2010 and 2022 in 59 LMICs. The Demographic and Health Survey is nationally representative cross-sectional survey conducted in LMICs that provides reliable data on women, men, and children. It measures key indicators that are considered important to allow countries to generate data that can be used to inform policy and practice. The DHS surveys use the same standardized data collection procedures, sampling, questionnaires, and coding, making the results comparable across countries. Therefore, datasets were appended together to explore factors associated with the number of ANC contacts among pregnant women in LMICs.

The survey employs a two-stage sampling procedure that involves the selection of census enumeration areas from each sampling stratum using a probability proportional to the size of the number of households in each enumeration area in the first stage to ensure the national representativeness. In the second stage, systematic random sampling was used sample households in each enumeration area to form survey clusters. A detailed description of the DHS sampling design and data collection procedures have been found in each country's DHS report. A total of 1,668,392 women of reproductive age in 59 LMICs were interviewed. However, this study was limited to married women aged 15–49 who had given birth within the last five years preceding the surveys. The total sample size was therefore 520,377, with the total number of respondents per country ranging from 1,194 in South Africa to 171,649 in India (**Table 1**).

### Study variables and measurement

The outcome variable for this study was the number of ANC contacts during the last pregnancy, measured as the discrete number of times the mother received ANC from the skilled providers.

**Table 1. Provides a descriptive summary of the survey country, year of survey, and sample size of all the countries included in the analyses.**

| Country | Year of survey | Sample size | Country | Year of survey | Sample size |
|---|---|---|---|---|---|
| Afghanistan | 2015 | 19,151 | Kyrgyz Republic | 2012 | 2,921 |
| Albania | 2017/18 | 2,289 | Lesotho | 2014 | 2,000 |
| Angola | 2015/16 | 6,346 | Liberia | 2019/20 | 2,930 |
| Armenia | 2015/16 | 1,357 | Madagascar | 2021 | 7,159 |
| Bangladesh | 2017/18 | 4,946 | Malawi | 2015/16 | 11,022 |
| Benin | 2017/18 | 8,150 | Maldives | 2016/17 | 2,168 |
| Burkina Faso | 2021 | 6,108 | Mali | 2018 | 5,960 |
| Burundi | 2016/17 | 7,633 | Mauritania | 2019/21 | 6,465 |
| Cambodia | 2021/22 | 4,277 | Mozambique | 2011 | 6.075 |
| Cameroon | 2018 | 5,014 | Myanmar | 2015/16 | 3,673 |
| Central Democratic Congo | 2013/14 | 9,503 | Namibia | 2013 | 1,543 |
| Chad | 2014/15 | 10,170 | Nepal | 2016 | 3,992 |
| Colombia | 2015 | 7,308 | Niger | 2012 | 7,431 |
| Comoros | 2012 | 1,624 | Nigeria | 2018 | 20,167 |
| Congo | 2011/12 | 5,024 | Pakistan | 2017/18 | 8,153 |
| Cote d'vore | 2021 | 4,805 | Papua New Guinea | 2016/18 | 5,582 |
| Dominican Republic | 2013 | 2,122 | Philippines | 2022 | 2,833 |
| Egypt | 2014 | 11,234 | Rwanda | 2019/20 | 5,030 |
| Ethiopia | 2016 | 6,645 | Senegal | 2010 | 7,471 |
| Gabon | 2012 | 2,857 | Sierra leone | 2019 | 5,448 |
| Gambia | 2019/20 | 5,385 | South Africa | 2016 | 1,194 |
| Ghana | 2014 | 3,658 | Tajikistan | 2017 | 4,086 |
| Guatemala | 2014/15 | 8,192 | Tanzania | 2022 | 4,813 |
| Guinea | 2018 | 5,045 | Timor Leste | 2016 | 4,804 |
| Haiti | 2016/17 | 4,268 | Togo | 2013 | 4,587 |
| Honduras | 2011/12 | 6,945 | Turkey | 2013 | 2,809 |
| India | 2019/21 | 171,649 | Uganda | 2016 | 8,399 |
| Indonesia | 2017 | 14,786 | Zambia | 2018 | 5,470 |
| Jordan | 2017/18 | 7,133 | Zimbabwe | 2015 | 4,061 |
| Kenya | 2022 | 8,386 | | | |

We examined independent variables at three levels (individual, household, and community-level factors). Level 1 included individual-level factors which were participants' responses to survey questions on the personal factors. Level 2 factors were characteristics of household from which the individuals came from. The individual level factors included age of women (15–24, 25–34, or ≥35 years), educational status of women (no education, primary, secondary, or higher), educational status of husband (no education, primary, secondary, higher, or don't know), planned pregnancy (yes or no), mass media exposure (yes or no), accessing health care (big problem and not big problem), working status (working or not working), timing of ANC (timely or delayed), birth order (1, 2–5, or >5), terminated pregnancy (yes or no) and women's decision making autonomy (yes or no). Household wealth status (poorest, poorer, middle, richer, or richest), number of household members (≤5, 5–10 or >10) and sex of household head (male or female) were included as household level variables. Level 3 included community-level factors such as place of residence (rural or urban).

Media exposure was generated from frequency of radio, television, and newspaper. It is categorized as "yes" if women who had exposure to at least one type of media; radio, newspaper, or television considered as and "no" otherwise.

The women's decision-making autonomy was generated from four areas of women's autonomy in decision making. These are person who usually decides on women's health care, person who usually decides on large household purchases, person who usually decides what to do with money partner earns and person who usually decides on visits to family or relatives. Then it was recoded as "yes" (women has decision-making autonomy) if responses are respondent alone, respondent and husband/partner or respondent and other person and "no" (does not have autonomy over decision-making) if responses are husband/partner alone, someone else, or others [31].

Accessing health care was generated from the DHS questions; getting the money needed for treatment (big problem/not a big problem), distance to a healthcare facility (big problem/not a big problem), having to take transport (big problem/not a big problem) and not wanting to go alone (big problem/not a big problem). It was coded as a binary variable as a big problem if a woman faces at least one above problem and not a big problem if a woman didn't report none of the above problem [32].

Timing of ANC visit was recorded as: "timely" if the first ANC visit was within 12 weeks of gestation "and "delayed" if women booked first ANC after 12 weeks of gestation.

## Data processing and analysis

The data cleaned and statistical analysis were carried out using STATA version 16. Frequencies and percentages were used to describe the background characteristics of the study participants. Since all variables were not complete, the regression analysis was based on a complete case analysis. We anticipate that using a complete case analysis did not bias our results because data availability for each DHS was determined by data collection at the time of the survey rather than the outcome. To identify associated factors of ANC, we used the Poisson regression model as a starting point because ANC contact (the dependent variable) is a nonnegative integer. To use Poisson regression, the assumption that the mean and variance are equal should be met. To handle overdisprsion and excess zeros, we have considered negative binomial Poisson regression and zero inflated Poisson models, respectively to obtain accurate results. The presence of overdisprsion and excess zeros in the data were compared by alpha test and vuong test, respectively.

The formula for Poisson regression and negative binomial regression models can be represented as:

$$\log(\mu) = \beta_0 + \beta_1 x_1 + \beta_2 x_2 + \cdots + \beta_k x_k$$

Where: log ($\mu$) is the natural logarithm of the mean of number of ANC contacts, $\beta_0$ is the overall intercept, $\beta_1, \beta_2, \ldots, \beta_k$ *are the regression coefficients, and* $x_1, x_2, \ldots, x_k$ are the independent variables.

However, the distribution of the outcome variable is different between the Poisson regression and the negative binomial regression model. The Poisson regression model assumes that the number of ANC contacts follows a Poisson distribution with mean (mean and variance of the number of ANC contacts are equal). On the other hand, the negative binomial regression assumes that the number of ANC contacts follows a negative binomial distribution with mean and additional parameter (r) which controls the variance independently of the mean:

$$P(Y = y) = \frac{\Gamma(y + r)}{\Gamma(y + 1) \cdot \Gamma(r)} \left(\frac{r}{r + \mu}\right)^r \left(\frac{\mu}{r + \mu}\right)^y$$

Where: $\mu$ is the mean of the number of ANC contacts, $r$ is the dispersion parameter, which controls the variance independently of the mean, and $\Gamma$ is the gamma function. When r = 1,

the negative binomial regression model becomes Poisson regression model without over-dispersion [33].

Zero-inflated Poisson regression is used when dealing with count data if zeros exceed the expected frequency of zeros predicted by the standard Poisson model. It involves two components: a logistic model that predicts excess zeros and a Poisson model that predicts count data [34].

The formula for zero-inflated Poisson regression can be represented as follows:

a. For the logistic component (modelling excess zeros):

$$\text{logit}\ (P(Y=0)) = \alpha_0 + \alpha_1 x_1 + \alpha_2 x_2 + \cdots + \alpha_k x_k$$

Where: $P(Y=0)$ is the probability of observing a zero count, logit is the logistic transformation, $\alpha_0$ is intercept, $\alpha_1, \alpha_2, \ldots, \alpha_k$ are the regression coefficients for the logistic component, and $x_1, x_2, \ldots, x_k$ are the predictor variables for the logistic component.

b. For the Poisson component (modelling the count data):

$$\log(\mu) = \beta_0 + \beta_1 x_1 + \beta_2 x_2 + \cdots + \beta_k x_k$$

Where: $\log(\mu)$ is the natural logarithm of the mean of number of ANC contacts, $\beta_0$ is the intercept, $\beta_1, \beta_2, \ldots, \beta_k$ are the regression coefficients for the Poisson component, and $x_1, x_2, \ldots, x_k$ are the independent variables for the Poisson component.

We employed multilevel models since DHS data are hierarchical, i.e., individuals were nested within communities. Using standard Poisson regression, negative binomial regression and zero inflated Poisson models could lead to biased estimates and misinterpretation of the results when the data is clustered/nested. A multilevel Poisson regression and negative binomial regression model is obtained by including cluster-specific random effects in the standard Poisson regression and negative binomial regression model. It can be represented as:

$$\log\left(\mu_{ij}\right) = \beta_0 + \beta_1 x_{ij1} + \beta_2 x_{ij2} + \cdots + \beta_k x_{ijk} + u_{0j}$$

Where: $\log(\mu_{ij})$ is the natural logarithm of the mean of the number of ANC contacts at the $i^{\text{th}}$ observation in the $j^{\text{th}}$ group, $\beta_0$ is the intercept, $\beta_1, \beta_2, \ldots, \beta_k$ are fixed effects coefficients for the individual-level predictors, $x_{ij1}, x_{ij2}, \ldots, x_{ijk}$ are the individual-level predictors, $u_{0j}$ is random intercept at the group level and $j$ indexes the different groups or clusters [35].

To cater for the unexplained variability at the community level, we used clusters as random effect. The log of the probability of the number of ANC was modelled using a three-level model. In particular, four models were constructed. First, we constructed a model containing the outcome variable only (null model) to decompose the total variance into its cluster and country components. Then model containing only individual-level variables (model I) and model containing individual-level and household level variables (model II) were fitted. Finally, in Model III, we adjusted for all individual, household and community-level variables to estimate the number of ANC and the factors. To select the best-fitted model Log likelihood test was used and the model with the highest Log likelihood was selected (model III). The Intraclass Correlation Coefficient (ICC), and proportional change in variance (PCV) were computed to assess the clustering effect/variability. ICC shows the variation in number of ANC contacts for women of reproductive age due to contextual characteristics and it was computed as follows [36, 37]: -

$$ICC = \frac{\text{the estimated variance of clusters in each model}}{\text{the estimated variance of clusters in each model } + 3.29}$$

Proportional change in variance was used to measure total variation attributed to individual and community level variables in the multilevel model (each model) as compared to the null model. It was calculated as: PCV % = (VA−VB/VA)*100; where: VA = variance of the null model, and VB = variance of the model with more factors [37].

First, we employed binary multilevel negative binomial regression model for each independent variable to select variables for multivariable analysis, and variables with p-value ≤ 0.20 in the binary multilevel negative binomial regression analysis were included in multivariable analysis. Finally, results for the multivariable analysis have been presented as incidence rate ratios (IRR), with their corresponding 95% confidence intervals (CI), and p-value <0.05 were considered to be significant factors associated with the number of ANC.

## Ethical consideration

Since this study used secondary data analysis of publicly available survey data, ethical approval is not required. However, to use the data we requested DHS Program, and we received an authentication letter from archive@dhsprogram.com.

## Results

### Background characteristics of study participants

A total of 520,377 women from 59 LMICs countries were included in the study. The mean age of study participants was 28.7 years, with a standard deviation of 6.5 years. Only 1 in 10 (10.2%) women and 10.4% of husbands had received higher education, and 29.4% of women and 27.1% of husbands had no formal education. The majority (70.1%) of study participants were exposed to at least one form of media (radio, newspaper or television) per week, and 76.9% of respondents had their first ANC visit within 12 weeks of gestation. Two-thirds, 157,623 (67.2%) of respondents had women's autonomy in decision making and more than two-thirds, 69.9% of respondents were rural residents (**Table 2**).

### Number of ANC contacts

The mean and standard deviation of ANC contacts are 4.8 and 4.2, respectively. One in ten (10.1%) of the pregnant women did not receive any ANC service. The proportion of pregnant women who received at least eight ANC visits was 16.9% (**Fig 1**).

### Selection of models

Poisson regression and negative binomial regression are tested to see over dispersion. As shown in the **Table 3**, the p-value for the alpha test is statistically significant (<0.001) which indicates that alpha is not zero (there is overdispersion). We then compare Poisson regression with zero-inflated Poisson regression to test for excess zeros. Vuong test was not statistically significant, indicating that excess zero is not a problem. Therefore, multilevel negative binomial regression was used to identify factors contributing to the number of ANC contacts. The random effect analysis in the empty model was used to examine the cluster effect on number of ANC. The results implied significant variability in ANC number across the clusters (ICC = 9.8%) which indicates that the cluster accounted for a 9.8% variance in ANC use. In

**Table 2. Background characteristics of study participants in LMICs, DHS 2010–2022.**

| Variables | Frequency | Percent | Missing values (%) |
|---|---|---|---|
| Maternal age | | | - |
| Mean+SD | 28.7+6.5 | | |
| 15–24 | 146,654 | 28.2 | |
| 25–34 | 269,630 | 51.8 | |
| 35–49 | 104,093 | 20.0 | |
| Women's education | | | |
| Not educated | 153,211 | 29.4 | |
| Primary | 123,035 | 23.6 | |
| Secondary | 191,191 | 36.8 | |
| Higher | 52,940 | 10.2 | |
| Wealth status | | | |
| Poorest | 129,863 | 25.0 | - |
| Poorer | 114,646 | 22.0 | |
| Middle | 102,844 | 19.8 | |
| Richer | 93,141 | 17.9 | |
| Richest | 79,883 | 15.3 | |
| Husband education | | | |
| Not educated | 99,453 | 27.1 | 153,112(29.4) |
| Primary | 94,580 | 25.7 | |
| Secondary | 126,794 | 34.5 | |
| Higher | 38,108 | 10.4 | |
| Don't know | 8,330 | 2.3 | |
| Working status | | | |
| Not working | 200,237 | 53.6 | 147,097(28.3) |
| Working | 173,043 | 46.4 | |
| Family size | | | |
| ≤5 | 238,614 | 45.9 | - |
| 5–10 | 227,942 | 43.8 | |
| >10 | 53,821 | 10.3 | |
| Parity | | | |
| primiparous | 128,246 | 24.6 | - |
| Multiparous | 283,517 | 54.5 | |
| Grand Multiparous | 108,614 | 20.9 | |
| Pregnancy intention | | | |
| Intended | 376,214 | 75.3 | 20,596(3.9) |
| Unintended | 123,567 | 24.7 | |
| Media exposure | | | |
| No | 152,609 | 29.9 | 10,632(2.0) |
| Yes | 357,136 | 70.1 | |
| Timing of ANC | | | |
| Timely | 359,502 | 76.9 | 52,734(10.1) |
| Delayed | 108,141 | 23.1 | |
| Birth order | | | |
| 1 | 128,246 | 24.6 | - |
| 2–5 | 321,853 | 61.9 | |
| >5 | 70,278 | 13.5 | |
| sex of household head | | | |

*(Continued)*

**Table 2.** (Continued)

| Variables | Frequency | Percent | Missing values (%) |
|---|---|---|---|
| Male | 446,499 | 85.8 | - |
| Female | 73,878 | 14.2 | |
| Residence | | | |
| Urban | 156,468 | 30.1 | - |
| Rural | 363,909 | 69.9 | |
| Accessing health care | | | |
| Big problem | 280,796 | 56.3 | 21,576(4.1) |
| Not big problem | 218,005 | 43.7 | |
| Women's decision-making autonomy | | | |
| Yes | 349,634 | 67.2 | 89(0.01) |
| No | 170,654 | 32.8 | |
| Terminated pregnancy | | | |
| No | 437,189 | 84.0 | 37(0.007) |
| Yes | 83,151 | 16.0 | |

Model III, the variability in ANC use decreased across clusters (ICC, 6.0%). In model I, model II, and model III, the explained variance was 25.0%, 41.7%, and 52.8% respectively. This means that a substantial amount of variances in the number of ANC contacts has been explained by model III. To identify factors associated with number of ANC, model III (full model) was selected as the most suitable due to the large likelihood ratio (p-value<0.001).

## Factors associated with number of ANC

The frequency of ANC was 1.07 (AIRR = 1.07, 95% CI: 1.06, 1.08), 1.19 (AIRR = 1.19, 95% CI: 1.17, 1.20), and 1.31 (AIRR = 1.31, 95% CI: 1.29, 1.32) times higher for women with primary, secondary, and higher education compared with women with no formal education respectively. Similarly, the frequency of ANC was 1.12 (AIRR = 1.12, 95% CI: 1.11, 1.13) and 1.11 (AIRR = 1.11, 95% CI: 1.09, 1.12) times higher for women whose husbands had secondary and higher education respectively. Women who access media had 34% higher ANC contacts when compared with women who do not access media (AIRR = 1.34, 95% CI: 1.32, 1.39). The frequency of ANC was 3.12 (AIRR = 3.12, 95% CI: 3.11, 3.14) times higher among women who had decision making autonomy compared to women who hadn't decision making autonomy. The frequency of ANC was 1.21 (AIRR = 1.21, 95% CI: 1.19, 1.24) and 1.32 (AIRR = 1.32, 95% CI: 1.29, 1.34) times higher for women from richer and richest households compared with women from poorest households respectively. Urban dweller women had 18% more ANC contacts compared with rural dweller women (AIRR = 1.18, 95% CI: 1.17, 1.20). Mothers aged 35–49 had 1.14 times more ANC contacts than mothers aged 15–24 (AIRR = 1.14, 95% CI: 1.13, 1.16). Moreover, women from female headed households had 45% more ANC contacts compared to women from male headed (AIRR = 1.45, 95% CI: 1.43, 1.46).

Women who said accessing healthcare was a big problem had 15% lower ANC contacts compared with women who said accessing healthcare was a not big problem (AIRR = 0.85, 95% CI: 0.83, 0.87). Women who initiated ANC after 12 weeks of gestation had 36% (AIRR = 0.64, 95% CI: 0.62, 0.65) lower ANC contacts as compared to those who initiated ANC within 12 weeks of gestation. Finally, grand multiparous women had lower number of ANC contacts (AIRR = 0.90, 95% CI = 0.89, 0.91) when compared with primiparous women (**Table 4**).

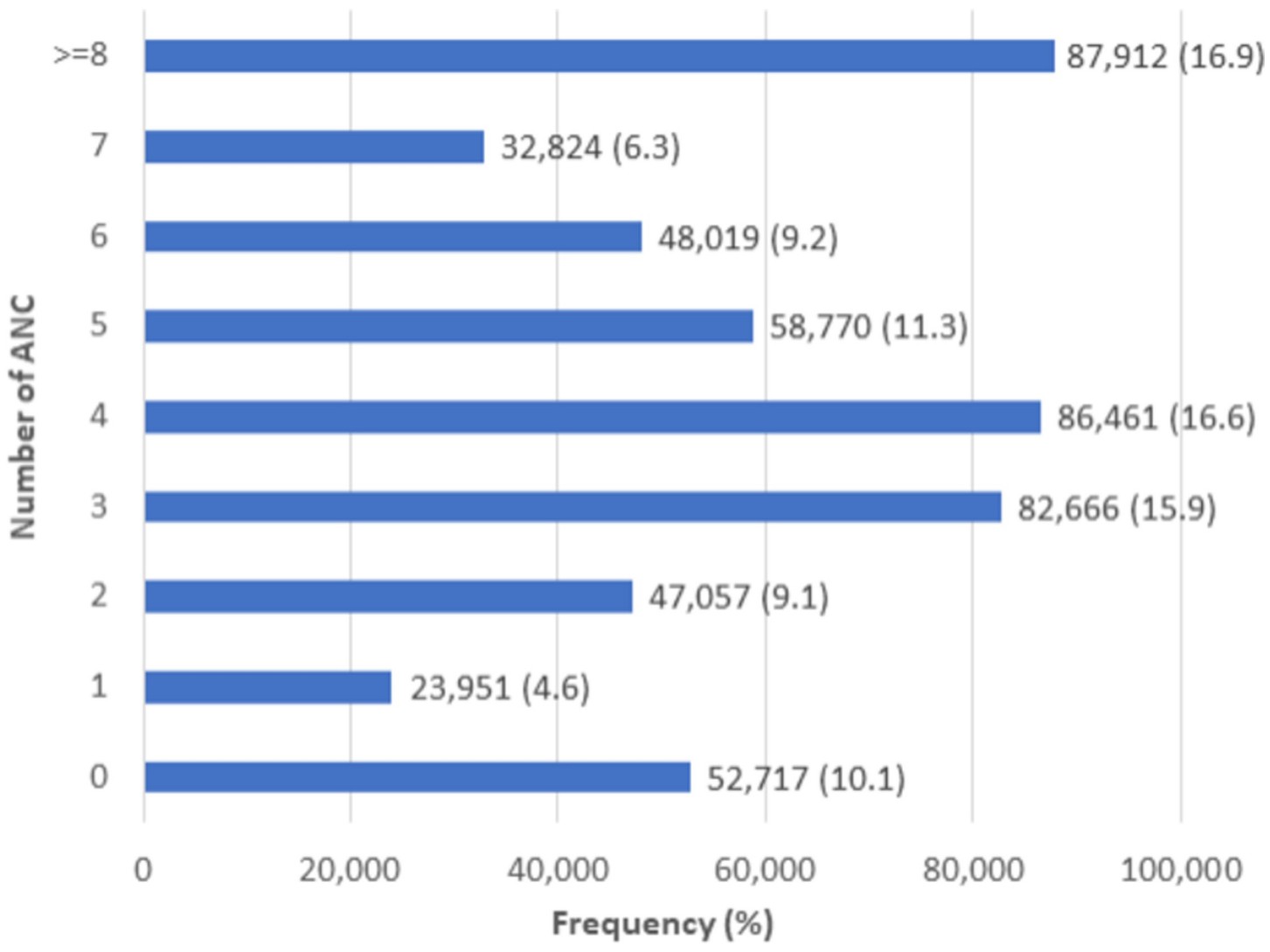

**Fig 1. Number of ANC contacts in low and middle-income countries, 2010–2022.**

**Table 3. Model selection procedure.**

| Poisson models | Log likelihood | | | | Tests |
|---|---|---|---|---|---|
| Poisson model | -1189393.6 | | | | |
| Negative binomial model | -9228932.9 | | | | Alpha = 0.090, p-value≤0.001 |
| Zero inflated model | -9202346.5 | | | | Vuong test, p-value = 0.112 |
| Multilevel models | Cluster variance | ICC (%) | PCV (%) | Log likelihood | Assumptions and Test |
| Null model | 0.36 | 9.8 | Reference | | |
| Model I | 0.27 | 7.6 | 25.0 | -512632.00 | Model I is nested within Model II |
| Model II | 0.21 | 6.0 | 41.7 | -512359.25 | LR chi2(7) = 545.49, p-value≤0.001 |
| Model III | 0.17 | 4.9 | 52.8 | -511810.04 | Model II is nested within Model III LR chi2(1) = 1098.43, p-value≤0.001 |

Note: ICC = intra-class correlation coefficient; Model I = model adjusted for individual-level factors; Model II = model adjusted for individual and household level factors; Model III = the full model adjusted for individual, household, and community-level factors; PCV = proportional change in variance

Table 4. Multi-level negative binomial regression on the number of antenatal care among women in LMICs, DHS 2010–2022.

| Variables | Model I AIRR (95% CI) | Model II AIRR (95% CI) | Model III AIRR (95% CI) |
|---|---|---|---|
| Maternal age | | | |
| 15–24 | 1 | 1 | 1 |
| 25–34 | 1.03 (1.00, 1.05) | 1.04(1.01, 1.06) | 1.08(1.07,1.09)* |
| 35–49 | 1.08(1.06, 1.09) | 1.09(1.07, 1.11) | 1.14(1.13, 1.16)** |
| Women's education | | | |
| Not educated | 1 | 1 | 1 |
| Primary | 1.04(1.02, 1.05) | 1.07(1.05, 1.09) | 1.07(1.06,1.08)** |
| Secondary | 1.09(1.04,1.11) | 1.12(1.09, 1.13) | 1.19(1.17,1.20)** |
| Higher | 1.19(1.18, 1.21) | 1.18(1.15, 1.20) | 1.31(1.29,1.32)** |
| Husband education | | | |
| Not educated | 1 | 1 | 1 |
| Primary | 1.08(1.05, 1.11) | 1.04(1.01, 1.10) | 1.07(1.06,1.08)* |
| Secondary | 1.21(1.20,1.24) | 1.19(1.16, 1.22) | 1.12(1.11,1.13)** |
| Higher | 1.09 (1.03, 1.12) | 1.13(1.10, 1.15) | 1.11(1.09,1.12)** |
| Don't know | 0.98(0.97,1.02) | 0.97 (0.96,1.01) | 0.99 (0.97,1.01) |
| Working status | | | |
| Not working | 1 | 1 | 1 |
| Working | 1.04(1.03,1.06) | 0.99(0.97, 1.01) | 1.01(0.99, 1.02) |
| Media exposure | | | |
| No | 1 | 1 | 1 |
| Yes | 1.39(1.38,1.40) | 1.32(1.30, 1.35) | 1.34(1.32,1.39)** |
| Timing of ANC | | | |
| Timely | 1 | 1 | 1 |
| Delayed | 0.71(0.69,0.73) | 0.68(0.67, 0.81) | 0.64(0.62,0.65)** |
| Accessing health care | | | |
| Not Big problem | 1 | 1 | 1 |
| Big problem | 0.89 (0.87,0.92) | 0.83(0.81,0.85) | 0.85(0.83,0.87)* |
| Women's decision-making autonomy | | | |
| No | 1 | 1 | 1 |
| Yes | 3.71(3.68,3.74) | 3.42(3.39,3.44) | 3.12(3.11,3.14)* |
| Terminated pregnancy | | | |
| No | 1 | 1 | 1 |
| Yes | 0.99(0.97, 1.02) | 1.02(0.99,1.04) | 1.03(0.99,1.04) |
| Parity | | | |
| Primiparous | 1 | 1 | 1 |
| Multiparous | 1.01 (0.99, 1.06) | 1.03(1.02, 1.06) | 0.98(0.95, 1.01) |
| Grand multiparous | 0.98(0.97, 1.03) | 1.02(0.99, 1.05) | 0.90(0.89, 0.91)* |
| Family size | | | |
| ≤5 | | 1 | 1 |
| 5–10 | | 0.98(0.96, 1.03) | 0.98(0.97, 1.01) |
| >10 | | 1.03(0.99, 1.04) | 0.91(0.90, 0.93)* |
| Wealth status | | | |
| Poorest | | 1 | 1 |
| Poorer | | 1.01(0.99, 1.03) | 1.02(0.99,1.03) |
| Middle | | 1.03(1.01, 1.06) | 1.01(0.99,1.02) |
| Richer | | 1.11(1.09, 1.14) | 1.21(1.19,1.24)** |

(*Continued*)

**Table 4.** (Continued)

| Variables | Model I AIRR (95% CI) | Model II AIRR (95% CI) | Model III AIRR (95% CI) |
|---|---|---|---|
| Richest | | 1.23(1.21, 1.25) | 1.32(1.29,1.34)** |
| Sex of household head | | | |
| Male | | 1 | 1 |
| Female | | 1.37(1.34,1.39) | 1.45(1.43, 1.46)* |
| Residence | | | |
| Rural | | | 1 |
| Urban | | | 1.18(1.17,1.20)** |

Note: AIRR; Adjusted incidence rate ratios; Model I = model adjusted for individual-level factors; Model II = model adjusted for individual and household level factors; Model III = the full model adjusted for individual, household, and community-level factors

* = p-value < 0.05

** = p-value < 0.001

## Discussion

This study investigated the factors associated with number of ANC among pregnant women in LMICs. It contributes to the literature on factors for number of ANC contacts considering count nature of ANC contacts. The current study identified several factors that affect number of ANC contacts.

In this study, women's educational attainment was statistically significant, with the number of ANC contacts increasing as women's educational attainment increased.. This finding is supported by previous studies [24, 26, 29]. Similarly, we found pregnant women whose partner's attained secondary and higher education are more likely to attain a higher number of ANC visits as compared to the uneducated category. This finding is supported by a study conducted in Ethiopia [26]. Moreover, studies conducted elsewhere found that positive association between mothers and their partner educational status and receiving adequate (completing ≥4 or ≥8) ANC contacts [18–20, 22, 25, 27]. This result could be because educated mothers and their partners have better access to information through media, thereby improving access to maternal health care and helping them understand the health benefits of receiving ANC services. Additionally, educated women are more likely to make decisions about their own health care and have greater economic social power than uneducated women [38, 39].

We found strong positive association between women decision making autonomy number of ANC. Thus, women with decision-making autonomy are more likely to have higher amounts of ANC. Likewise, previous research has shown that women with autonomy in healthcare decision-making are more likely to have a greater number of ANC contacts [19, 40–42]. The strong association between mothers' autonomy and utilization of ANC indicates that interventions designed to empower women may increase their use of maternal health care services and gender perspective should be considered reproductive health development efforts.

We found that mothers from wealthier and wealthiest families were more likely to have a higher number of ANC contacts than mothers from the poorest families. This finding is supported by previous studies [18–20, 22, 25–27, 29], which revealed that mothers from poorest households were fewer number of ANC contacts than those from other categories. Moreover, as observed in previous studies [19, 20, 22, 25, 27], our study showed that women who perceived access to healthcare to be a big problem had a lower number of ANC contacts compared with women who perceived access to healthcare to be a not a big problem. Although ANC services are provided free of charge to all mothers, there are direct and indirect costs associated

with attending maternity care services [43]. Therefore, pregnant women from households with better economic status can pay direct health service costs like transportation costs and indirect costs like household and work obligations, while mothers from poorest households need financial capacity to support their daily living and therefore, they may be spent more time on economic activities to cater to their families rather than their health. Besides, mothers from wealthy families have access to information about maternal healthcare through mass media [44]. In contrast study conducted in Nigeria shows that mothers from poorer households had 52% lower odds of use ANC compared to mothers from poorest household [21].

However, risks of maternal and fetal complications are common among adolescent women [45, 46], we observed number of ANC contacts was lower among mothers aged 15–24 years. This result is consistent with some previous research conducted elsewhere showing that the number of ANC contacts increases as maternal age increases [18, 24, 26]. The justification for this result might be that younger women are less likely to recognize their pregnancy early [47]. Study shows women may book ANC appointments earlier if pregnancy is detected early [48]. This study and previous studies conducted in Nigeria [21] indicate that early ANC booking time is one of the important factors influencing the number of ANC contacts. However, a study conducted in Ethiopia and Guinea showed that mothers aged 35–49 had a lower number of ANC contacts compared with mothers aged 15–24 years [19, 25]. In addition, a study conducted in SSA countries found that women aged 35 years and older had lower odds of partial utilization of ANC (1–3 contacts) compared with women aged 15–24 years. However, similar study reported that the higher odds of receiving adequate ANC (≥4 contacts) women aged 25–34 years and ≥35 years compared to women aged 15–24 [27]. This disagreement might be due to the differences in the study population and methods of analysis used. For example, unlike the current study, which was conducted on both rural and urban mothers (nationally representative data), the study conducted by Hassen SS et.al was limited to the rural mothers [25]. Furthermore, unlike previous studies [19, 27], this study was limited to married mothers in order to incorporate partner-related factors and use count models to obtain optimal estimates.

Furthermore, our results showed that women with media exposure received a higher number of ANC contacts during pregnancy compared to women with no media exposure, consistent with research conducted in Ethiopia [25, 29], Nigeria [21], and SSA countries [20, 27]. Women exposed to the media may have better awareness and understanding of t the existence of maternal health care services and the importance of healthcare utilization. Mothers' attitude towards maternal care is crucial to her ability to access the requisite care since behavioral modification is a challenging task in terms of healthcare. Globally, mass media is one of the popular and effective tools for health promotion and behavioral change especially LMICs [49–52]. Mass media is used as tool to disseminate maternal healthcare information and enable mothers/communities shape their health beliefs and perceptions [53, 54]. Therefore, mothers who receive information about maternal healthcare through mass media are expected to be able to make better health decisions by understanding risk factors, preventive measures, and availability of professional services. In addition to serving as a source of information on maternal health care utilization, mass media play a key role in reducing population use of tobacco, alcohol, and drugs, promoting cancer screening, and reducing HIV infection rates [55–57].

Like a study conducted elsewhere [20–22, 24, 27, 29], this study found that urban mothers are more likely to have a higher number of ANC contacts than rural mothers. The lower ANC contacts among rural dwellers is likely to be attributed to the lack of adequate health facilities/ long distances from health facilities to get services, lack of skilled health care providers, lack of infrastructure (e.g. road conditions, costs of transport), and lack of health information [32, 58, 59]. Another possible explanation is that rural residents may have lower mass media exposure

and education levels than urban residents. As majority of mothers live in rural areas, increased efforts are expected to increase the number of ANC at national level through home visits, outreach programs and mass media campaigns [57].

The results of this study are based on data collected following internationally accepted standard acceptable methodological procedures and on a large sample size covering 59 LMICs. Furthermore, since the number of ANC is count data, the results of this study are based on count modelling. Therefore, results are representative and applicable to pregnant mothers in LMICs. Despite the strength presented here, the result cannot show causal effect relation because the data were collected based on cross-sectional study design. Due to the secondary nature of data, the result did not include all factors that may affect outcome like knowledge, attitude, and satisfaction of mothers on ANC services. Furthermore, these data depend on maternal memory and may therefore be subject to recall bias. Finally, our regression model coefficients were unweighted and may therefore underestimate the standard errors of coefficients.

## Conclusion

This study revealed that being older age group, higher maternal/partners education, mothers from households with better wealth, improved household wealth status; mothers having access to mass media, mothers who had decision making autonomy, urban dwellers, and mother with more than one birth order were significantly associated with more number of ANC contacts. However, mothers who said accessing healthcare was a big problem and mothers who initiated ANC after 12 weeks of gestation were significantly associated with less number of ANC contacts. This study suggests policymakers and programmers to prioritize their programs and plan for rural mothers, uneducated mothers, and their partners to increase number of ANC contacts to enhance maternally and newborn health. Besides, improving community awareness about maternal health care services through mass media, community dialoging, and health extension program should be areas of priority for policymakers.

## Acknowledgments

The author would like to acknowledge Demographic Health and Survey (DHS) program managers, which granted us the permission to use DHS data for this study.

## Author Contributions

**Conceptualization:** Adugnaw Zeleke Alem.

**Data curation:** Adugnaw Zeleke Alem, Biresaw Ayen Tegegne, Fantu Mamo Aragaw, Rediet Eristu Teklu, Tsegaw Amare Baykeda.

**Formal analysis:** Adugnaw Zeleke Alem, Biresaw Ayen Tegegne, Fantu Mamo Aragaw, Rediet Eristu Teklu, Tsegaw Amare Baykeda.

**Investigation:** Adugnaw Zeleke Alem, Biresaw Ayen Tegegne, Fantu Mamo Aragaw, Rediet Eristu Teklu, Tsegaw Amare Baykeda.

**Methodology:** Adugnaw Zeleke Alem, Biresaw Ayen Tegegne, Fantu Mamo Aragaw, Rediet Eristu Teklu, Tsegaw Amare Baykeda.

**Software:** Biresaw Ayen Tegegne.

**Supervision:** Biresaw Ayen Tegegne, Fantu Mamo Aragaw, Rediet Eristu Teklu, Tsegaw Amare Baykeda.

**Validation:** Adugnaw Zeleke Alem, Biresaw Ayen Tegegne, Fantu Mamo Aragaw, Rediet Eristu Teklu, Tsegaw Amare Baykeda.

**Visualization:** Rediet Eristu Teklu.

**Writing – original draft:** Adugnaw Zeleke Alem, Biresaw Ayen Tegegne, Fantu Mamo Aragaw, Rediet Eristu Teklu, Tsegaw Amare Baykeda.

**Writing – review & editing:** Adugnaw Zeleke Alem, Biresaw Ayen Tegegne, Fantu Mamo Aragaw, Rediet Eristu Teklu, Tsegaw Amare Baykeda.

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
