## [Decision Letter · Decision Letter 0]

13 Jun 2023

PONE-D-22-24935Multilevel negative binomial analysis of factors associated with numbers of antenatal care contacts in low and middle income countries: Findings from 56 nationally representative dataPLOS ONE

Dear Dr. Alem,

Thank you for submitting your manuscript to PLOS ONE. After careful consideration, we feel that it has merit but does not fully meet PLOS ONE’s publication criteria as it currently stands. Therefore, we invite you to submit a revised version of the manuscript that addresses the points raised during the review process.

Please note that we have only been able to secure a single reviewer to assess your manuscript. We are issuing a decision on your manuscript at this point to prevent further delays in the evaluation of your manuscript. Please be aware that the editor who handles your revised manuscript might find it necessary to invite additional reviewers to assess this work once the revised manuscript is submitted. However, we will aim to proceed on the basis of this single review if possible. Your manuscript has been assessed by an expert reviewer, whose comments are appended below. The reviewer has highlighted concerns about several aspects of the methodology and the clarity of some of the scientific writing. Please ensure you respond to each point carefully in your response to reviewers document, and modify your manuscript accordingly.

We look forward to receiving your revised manuscript.

Kind regards,

Dr Joseph Donlan

Senior Editor

PLOS ONE

“The authors have not declared a specific grant for this research from any funding agency in the public, commercial or not-for-profit sectors.”

Reviewers' comments:

Reviewer's Responses to Questions

**Comments to the Author**

1. Is the manuscript technically sound, and do the data support the conclusions?

Reviewer #1: Yes

2. Has the statistical analysis been performed appropriately and rigorously? 

Reviewer #1: No

3. Have the authors made all data underlying the findings in their manuscript fully available?

Reviewer #1: No

4. Is the manuscript presented in an intelligible fashion and written in standard English?

Reviewer #1: No

5. Review Comments to the Author

Reviewer #1: Thank you so much for allowing me to revise the article.

General comments

1. line # 2 and 3 include sets. Instead of Findings from 56 nationally representative data. Better “Findings from 56 nationally representative data sets”.

2. Line #33# why you employed multilevel negative binomial regression, why not poison regression?

3. Why you focused only on individual variables? Why not include community level variables?

4. Line # 45# conclusions section: your recommendation is general beater specific

5. Line # 53# Keywords: Better focused on your title, what was Count modeling?

6. Line #67-69# re write the statement

7. Line #70-71# “Development Goals (SDGs) aimed to reduce maternal mortality (MMR) to 70 per 100,000 live births by 2030”. Please include globally

8. Line # 78# instead of still birth, better to say still births.

9. Line # 92 # is not clear, what does it mean “54% of maternal 95 deaths and 33% of stillbirths in LMICs”

10. Line # 140-141# is your data sets include all low and middle-income countries and what about the sample size is it weighted or un weighted?

11. Line #198-199# please briefly discuses the Intra-class Correlation Coefficient (ICC) and proportional change in variance (PCV)?

6. PLOS authors have the option to publish the peer review history of their article (what does this mean?). If published, this will include your full peer review and any attached files.

Reviewer #1: No

---

## [Author Response · Author response to Decision Letter 0]

26 Jul 2023

Subject: Point by point response for reviewer 

Title: Multilevel negative binomial analysis of factors associated with numbers of antenatal care contacts in low and middle income countries: Findings from 56 nationally representative data sets 

Manuscript ID: PONE-D-22-24935

Dear editor and reviewer, 

First and foremost, we would like to express our gratitude for giving us those very critical and lesson-giving comments and suggestions to improve our paper. We have addressed all the reviewer suggestions and comments to the manuscript. Our point-by-point responses for each comment and question are described in detail on the following pages. In addition, details of the changes are shown through Track Changes in the accompanying Supplementary File.

Editor comment: Thank you for stating the following financial disclosure:

“The authors have not declared a specific grant for this research from any funding agency in the public, commercial or not-for-profit sectors.”

Response: Thank you. We have amended funding statement as “The authors received no specific funding for this work” in the cover letter and revised manuscript.

Reviewer comments: General comments 

1. line # 2 and 3 include sets. Instead of Findings from 56 nationally representative data. Better “Findings from 56 nationally representative data sets”. 

Response: Thank you. We have replaced “Findings from 56 nationally representative data” with “Findings from 56 nationally representative data sets” in the revised manuscript.

2. Line #33# why you employed multilevel negative binomial regression, why not poison regression?

Response: Thank you for your concern. As we have stated in methods section (line 181-185), to run Poisson regression, the assumption of mean and variance equality should be satisfied. To handle overdisprsion, we have considered negative binomial Poisson regression to get precise results. We have tested presence of overdisprsion in the data by using alpha test under the following hypothesis: -

the null hypothesis (Ho): alpha is zero (there is no overdisprsion) and

 alternative hypothesis (Ha): alpha≠0 (there is overdisprsion). As shown in table 3 p value for this hypothesis was statistically significant (i.e. Ho was rejected). If there is overdispersion of data, a negative binomial model fits better. Therefore, we have employed multilevel negative binomial regression instead of poison regression. 

3. Why you focused only on individual variables? Why not include community level variables? 

Response: Thank you. This study uses all available variables in demographic and health survey. Of included variables, most of them were individual level variables and some of the were household level variables (Sex of household head, family size and wealth status), and community level variables (residence). As mentioned in methods section, models were constructed by adding household level variables and community level variables (residence) on individual level variables. Aggregate variables are not included in the analysis because aggerating variables from individual level variables have its own disadvantage. One of them is possibility of measurement error and endogeneity in the aggregate variables, which can bias the estimates of their effects on the individual outcomes. E.g., if when we use the mean of individual characteristics (such as education, income, etc.) to measure the community level variables, we may introduce error and correlation with individual level variables that can distort the results. 

4. Line # 45# conclusions section: your recommendation is general beater specific

Response: Thank you. We have modified accordingly in the revised manuscript.

5. Line # 53# Keywords: Better focused on your title, what was Count modeling?

Response: Thank you. We have replaced “Count modeling” with “Multilevel negative binomial analysis” in the revised manuscript.

6. Line #67-69# re write the statement

Response: Thank you. We have re write accordingly (i.e., we have deleted “with good ANC quality care, the majority of maternal deaths, neonatal deaths, and stillbirths can be prevented” because advantages of ANC is already stated in paragraph 2).

7. Line #70-71# “Development Goals (SDGs) aimed to reduce maternal mortality (MMR) to 70 per 100,000 live births by 2030”. Please include globally.

Response: Thank you. We have provided “globally” after 2030.

8. Line # 78# instead of still birth, better to say still births.

Response: Thank you. We have replaced “still birth” with “still births”.

9. Line # 92 # is not clear, what does it mean “54% of maternal 95 deaths and 33% of stillbirths in LMICs”

Response: Thank you. We have revised this sentence accordingly (we have deleted 95) in the revised manuscript. 

10. Line # 140-141# is your data sets include all low and middle-income countries and what about the sample size is it weighted or un weighted?

Response: Thank you. All low- and middle-income countries that have conducted standard demographic and health surveys between 2010 and 2020 are included in the analysis. Sample size is weighted. 

11. Line #198-199# please briefly discuses the Intra-class Correlation Coefficient (ICC) and proportional change in variance (PCV)?

Response: Thank you. We have elaborated it in the revised manuscript (methods section, page 11, line 195-204).

---

## [Decision Letter · Decision Letter 1]

6 Nov 2023

PONE-D-22-24935R1Multilevel negative binomial analysis of factors associated with numbers of antenatal care contacts in low and middle income countries: Findings from 56 nationally representative data setsPLOS ONE

Dear Dr. Alem,

Thank you for submitting your manuscript to PLOS ONE. After careful consideration, we feel that it has merit but does not fully meet PLOS ONE’s publication criteria as it currently stands. Therefore, we invite you to submit a revised version of the manuscript that addresses the points raised during the review process.

We look forward to receiving your revised manuscript.

Kind regards,

Manisha Nair, DPhil, MSc, MBBS

Academic Editor

PLOS ONE

Journal Requirements:

**Additional Editor Comments:**

Explain clearly why only 56 countries were included and why others like India were left out. The justification that only countries with DHS data between 2010 and 2020 were included do not hold as countries like India and Sri Lanka had surveys within this time-period. Please explain why these and other such countries were left out. Also arrange Table-1 in alphabetical order of the name of the countries

Please edit the following sentence “A total of 912,062 reproductive-age (15-49 years) were interviewed in 56 LMICs.” Do you mean women of reproductive age group?

Please edit the following sentence “Media exposure was generated from women’s asked on frequency of radio, ….”. What do you mean by “women’s asked on”?

In this sentence, do you mean ‘not a big problem’ (if a woman didn’t report none of the above problem were considered as no a big problem)?

You do not need to write ‘times’ after the % in the following sentences –

big problem had 12% times lower ANC contacts

ANC after 12 weeks of gestation had 18% (IRR=0.82, 95% CI: 0.81, 0.83) times lower ANC contacts

There is a need to carefully edit out all grammatical typing errors in the manuscript. There are a lot of them.

Reviewers' comments:

Reviewer's Responses to Questions

**Comments to the Author**

1. If the authors have adequately addressed your comments raised in a previous round of review and you feel that this manuscript is now acceptable for publication, you may indicate that here to bypass the “Comments to the Author” section, enter your conflict of interest statement in the “Confidential to Editor” section, and submit your "Accept" recommendation.

Reviewer #2: All comments have been addressed

Reviewer #3: All comments have been addressed

2. Is the manuscript technically sound, and do the data support the conclusions?

Reviewer #2: Partly

Reviewer #3: Yes

3. Has the statistical analysis been performed appropriately and rigorously? 

Reviewer #2: Yes

Reviewer #3: Yes

4. Have the authors made all data underlying the findings in their manuscript fully available?

Reviewer #2: No

Reviewer #3: Yes

5. Is the manuscript presented in an intelligible fashion and written in standard English?

Reviewer #2: Yes

Reviewer #3: Yes

6. Review Comments to the Author

Reviewer #2: The researcher must point out the Sustainable Development Goals and maternal mortality

The researchers did not mention the reference they relied on to apply the methodology or the package used in STATA program

The researchers did not mention the details of the model used, the form of the equation, and its statistical assumptions

In presenting the results, researchers should focus on commenting on percentages and not mention both numbers and percentages at the same time in sentences.

The researchers overlooked an important variable, which is the number of previous children the woman had given birth to. The more children a woman has, the less likely she is to receive prenatal care during pregnancy, as the woman becomes more experienced and less afraid of pregnancy. Why did researchers exclude this variable?

At the head of Table 4, researchers should differentiate between IRR and AIRR, You must explain the difference between them

Researchers should avoid large paragraphs in the discussion, and the discussion section should be shortened because it is very long

Reviewer #3: 1. Key words should be precise, not a combination of four or five word.

2. In abstract part write some finding in words, do not use numbers always.

3. The policy suggestion in abstract is not particular at all, authors need to specify who should be the authority for policy interventions.

4. Why the study undertakes 56 countries for the study? Yes, I understand it is for LMIC’s. But who should be responsible for policy implications? As countries are located at different continents. Please write rationale behind the study in one or two sentence.

5. In discussion part many of the articles used as “Previous studies”, are studies conducted in India and some articles from developed countries. But, the 56 countries for the study, either does not contain India, nor there is ant developed country undertaken.

6. I suggest to remove such studies as the economic, political, social, and cultural setup are varies across nations and regions.

7. PLOS authors have the option to publish the peer review history of their article (what does this mean?). If published, this will include your full peer review and any attached files.

Reviewer #2: No

Reviewer #3: **Yes: **Tushar Dakua

---

## [Author Response · Author response to Decision Letter 1]

13 Jan 2024

Subject: Point by point response for editor and reviewers 

Title: Multilevel negative binomial analysis of factors associated with numbers of antenatal care contacts in low and middle income countries: Findings from 59 nationally representative datasets

Manuscript ID: PONE-D-22-24935R1

Dear editor and reviewers, 

First and foremost we would like to express our gratitude for giving us those very critical and lesson-giving comments and suggestions to improve our paper. We have addressed all the reviewers' suggestions and comments to the manuscript. Our point-by-point responses for each comment and question are described in detail on the following pages. In addition, details of the changes are shown through Track Changes in the accompanying File.

Comments and authors response

Journal Requirements:

Please review your reference list to ensure that it is complete and correct. If you have cited papers that have been retracted, please include the rationale for doing so in the manuscript text or remove these references and replace them with relevant current references. Any changes to the reference list should be mentioned in the rebuttal letter that accompanies your revised manuscript. If you need to cite a retracted article, indicate the article’s retracted status in the References list and also include a citation and full reference for the retraction notice.

Response: Thank you. We have revised completeness and correctness of references accordingly. For example, references 7, 9, 11, 18, 23, and others references completeness and correctness were checked.

 Explain clearly why only 56 countries were included and why others like India were left out. The justification that only countries with DHS data between 2010 and 2020 were included do not hold as countries like India and Sri Lanka had surveys within this time-period. Please explain why these and other such countries were left out. Also arrange Table-1 in alphabetical order of the name of the countries

Response: Thank you for your concern: We have included India and other two countries (Madagascar and Mauritania) which were left out previously and update dataset for Burkina Faso, Nepal, Cambodia Cote d’vore, Philippines, Tanzania, and Kenya in the revised manuscript. We are aware of Sri Lanka has 2016 demographic and health survey, however, we haven’t found Sri Lanka’s 2016 demographic and health survey at archive@dhsprogram.com website. Moreover, we have arranged Table 1 alphabetically.

Please edit the following sentence “A total of 912,062 reproductive-age (15-49 years) were interviewed in 56 LMICs.” Do you mean women of reproductive age group?

Response: Thank you. We have revised it (methods, page 6, line 122-23).

Please edit the following sentence “Media exposure was generated from women’s asked on frequency of radio, ….”. What do you mean by “women’s asked on”?

Response: Thank you. We have revised it accordingly (methods, page 8, line 147-49).

In this sentence, do you mean ‘not a big problem’ (if a woman didn’t report none of the above problem were considered as no a big problem)?

Response: Thank you. We have revised it (methods, page 8, line 162-63).

You do not need to write ‘times’ after the % in the following sentences –

big problem had 12% times lower ANC contacts

ANC after 12 weeks of gestation had 18% (IRR=0.82, 95% CI: 0.81, 0.83) times lower ANC contacts

Response: Thank you. We have deleted times after the %.

There is a need to carefully edit out all grammatical typing errors in the manuscript. There are a lot of them.

Response: Thank you. We have carefully revised grammatical errors in the revised manuscript.

Reviewer #2: 

The researcher must point out the Sustainable Development Goals and maternal mortality

Response: Thank you. The Sustainable Development Goals and maternal mortality are described in background section (page 3, line 56-64). 

The researchers did not mention the reference they relied on to apply the methodology or the package used in STATA program

Response: Thank you. We have provided references accordingly. 

The researchers did not mention the details of the model used, the form of the equation, and its statistical assumptions

Response: Thank you. We have provided details of the model used, the form of the equation, and its statistical assumptions in the revised manuscript (methods, page 9-11, line 176-220).

In presenting the results, researchers should focus on commenting on percentages and not mention both numbers and percentages at the same time in sentences.

Response: Thank you. Only percentages are used to describe respondents’ characteristics (results, page 12&13, line 252-58).

The researchers overlooked an important variable, which is the number of previous children the woman had given birth to. The more children a woman has, the less likely she is to receive prenatal care during pregnancy, as the woman becomes more experienced and less afraid of pregnancy. Why did researchers exclude this variable?

Response: Thank you. We included birth order in previous analyses, which is similar to parity (the number of children a woman has had previously) in our case (study population). However, to make more clearer to readers and to use meaningful categories, we have replaced birth order with parity in the revised manuscript.

At the head of Table 4, researchers should differentiate between IRR and AIRR, You must explain the difference between them

Response: Thank you. IRR indicates crude incidence rate ratio and AIRR indicates adjusted incidence rate ratio. To assess factors associated with the number of ANCs, we used the AIRR, an epidemiologically relevant effect measure that adjusts for all important confounders. Sorry for our previous mistake, we have now replaced IRR with AIRR.

Researchers should avoid large paragraphs in the discussion, and the discussion section should be shortened because it is very long

Response: Thank you. We have revised accordingly (one paragraph includes single idea).

Reviewer #3:

1. Key words should be precise, not a combination of four or five word.

Response: Thank you. We have revised key words in the revised manuscript.

2. In abstract part write some finding in words, do not use numbers always.

Response: Thank you. We have used words to explain our findings in abstract section (page 2, line 37-42).

3. The policy suggestion in abstract is not particular at all, authors need to specify who should be the authority for policy interventions.

Response: Thank you. Now we revised it (page 2, line 44-48).

4. Why the study undertakes 56 countries for the study? Yes, I understand it is for LMIC’s. But who should be responsible for policy implications? As countries are located at different continents. Please write rationale behind the study in one or two sentence.

Response: Thank you. We conducted this study in low- and middle-income countries to increase the power of the study because, as shown in the background section, there are conflicting findings regarding factors in antenatal care. Low and middle income countries use similar WHO guidelines for antenatal care. Therefore, local stakeholders and international stakeholders such as the World Health Organization should take responsibility for antenatal care.

5. In discussion part many of the articles used as “Previous studies”, are studies conducted in India and some articles from developed countries. But, the 56 countries for the study, either does not contain India, nor there is ant developed country undertaken.

Response: Thank you. We have included India in the revised manuscript.

6. I suggest to remove such studies as the economic, political, social, and cultural setup are varies across nations and regions.

Response: Thank you. We have included India in the revised manuscript and only one reference from developed countries have been used to support general facts.

---

## [Editor Report · Decision Letter 2]

16 Feb 2024

PONE-D-22-24935R2Multilevel negative binomial analysis of factors associated with numbers of antenatal care contacts in low and middle income countries: Findings from 59 nationally representative datasetsPLOS ONE

Dear Dr. Alem,

Thank you for submitting your manuscript to PLOS ONE. After careful consideration, we feel that it has merit but does not fully meet PLOS ONE’s publication criteria as it currently stands. Therefore, we invite you to submit a revised version of the manuscript that addresses the points raised during the review process.

The authors have adequately addressed reviewers' comments. Please address the following minor points:1. Were survey weights used in the regression models? If so, how were they used and if not used, please explain why not used. Typically survey weights from DHS data should be used and a few sentences should be written about how the weights were handled in the pooled dataset.2. The discussion section needs to be shorter and please correct grammatical errors and typos.3. Please start the results section by describing the mean age of participants, education etc., as in Table 2. It abruptly starts with exposure to media.4. There is nothing written about missing data. Please show missing data for each variable in Table 2 and explain how missing data were handled.

We look forward to receiving your revised manuscript.

Kind regards,

Manisha Nair, DPhil, MSc, MBBS

Academic Editor

PLOS ONE
---

## [Author Response · Author response to Decision Letter 2]

13 Mar 2024

PONE-D-22-24935R2

Multilevel negative binomial analysis of factors associated with numbers of antenatal care contacts in low and middle income countries: Findings from 59 nationally representative datasets

The authors have adequately addressed reviewers' comments. Please address the following minor points:

1. Were survey weights used in the regression models? If so, how were they used and if not used, please explain why not used. Typically survey weights from DHS data should be used and a few sentences should be written about how the weights were handled in the pooled dataset.

Author response: We appreciate your concern. As Demographic and Health Surveys (DHS) employed a stratified, two-stage cluster sampling design with oversampling in certain population subgroups, weighting of DHS data is vital to correct for the unequal probabilities of selection, and to adjust the sample to be representative of the population. Pooled DHS data sets are often weighted using complex survey weights (sampling weights, primary sampling units, and stratification). However, the multilevel negative binomial regression does not directly support survey weights like other regression models. Due to the nature of the regression model, the regression coefficients we used were unweighted. In the revised manuscript, we acknowledge this issue (discussion section, line 414-16). 

2. The discussion section needs to be shorter and please correct grammatical errors and typos.

Author response: Thank you. We understand that our discussion section is long since we have many significant factors to be discussed. In the revised manuscript, we have removed duplicated ideas and extensively edited grammatical errors and typos.

3. Please start the results section by describing the mean age of participants, education etc., as in Table 2. It abruptly starts with exposure to media.

Author response: Thank you. We have stared describing mean of study participants (results section, line 256-259).

4. There is nothing written about missing data. Please show missing data for each variable in Table 2 and explain how missing data were handled.

Author response: Thank you. We have provided missing values for each variable in table 2 (column 4). As we have used Stata for statistical analysis which is robust model to handle missing value using complete case analysis, our regression model is based on complete case analysis. Therefore, we have explained missing data management in the revised manuscript (methods section, line 170-73).

---

## [Editor Report · Decision Letter 3]

18 Mar 2024

Multilevel negative binomial analysis of factors associated with numbers of antenatal care contacts in low and middle income countries: Findings from 59 nationally representative datasets

PONE-D-22-24935R3

Dear Dr. Alem,

We’re pleased to inform you that your manuscript has been judged scientifically suitable for publication and will be formally accepted for publication once it meets all outstanding technical requirements.

Kind regards,

Manisha Nair, DPhil, MSc, MBBS

Academic Editor

PLOS ONE

Additional Editor Comments (optional):

The authors have adequately addressed all comments.
---

## [Editor Report · Acceptance letter]

21 Mar 2024

PONE-D-22-24935R3 

PLOS ONE

Dear Dr. Alem, 

I'm pleased to inform you that your manuscript has been deemed suitable for publication in PLOS ONE. Congratulations! Your manuscript is now being handed over to our production team.

Kind regards, 

on behalf of

Associate Professor Manisha Nair 

Academic Editor

PLOS ONE